# Economic and Performance Evaluation of E-Health before and after the Pandemic Era: A Literature Review and Future Perspectives

**DOI:** 10.3390/ijerph20054038

**Published:** 2023-02-24

**Authors:** Helena Biancuzzi, Francesca Dal Mas, Chiara Bidoli, Veronica Pegoraro, Maristella Zantedeschi, Pietro Antonio Negro, Stefano Campostrini, Lorenzo Cobianchi

**Affiliations:** 1Department of Economics, Ca’ Foscari University of Venice, 30123 Venice, Italy; 2Department of Management, Ca’ Foscari University of Venice, 30123 Venice, Italy; 3Department of Clinical, Diagnostic and Pediatric Sciences, University of Pavia, 27100 Pavia, Italy; 4General Surgery Department, IRCCS Policlinico San Matteo Foundation, 27100 Pavia, Italy; 5ITIR—Institute for Transformative Innovation Research, 27100 Pavia, Italy

**Keywords:** e-Health, performance measurements, literature review, telemedicine, economic evaluation

## Abstract

E-Health represents one of the pillars of the modern healthcare system and a strategy involving the use of digital and telemedicine tools to provide assistance to an increasing number of patients, reducing, at the same time, healthcare costs. Measuring and understanding the economic value and performance of e-Health tools is, therefore, essential to understanding the outcome and best uses of such technologies. The aim of this paper is to determine the most frequently used methods for measuring the economic value and the performance of services in the framework of e-Health, considering different pathologies. An in-depth analysis of 20 recent articles, rigorously selected from more than 5000 contributions, underlines a great interest from the clinical community in economic and performance-related topics. Several diseases are the object of detailed clinical trials and protocols, leading to various economic outcomes, especially in the COVID-19 post-pandemic era. Many e-Health tools are mentioned in the studies, especially those that appear more frequently in people’s lives outside of the clinical setting, such as apps and web portals, which allow for clinicians to keep in contact with their patients. While such e-Health tools and programs are increasingly studied from practical perspectives, such as in the case of Virtual Hospital frameworks, there is a lack of consensus regarding the recommended models to map and report their economic outcomes and performance. More investigations and guidelines by scientific societies are advised to understand the potential and path of such an evolving and promising phenomenon.

## 1. Introduction

The World Health Organization (WHO) estimates that life expectancy has increased by six years over the past two decades. At a global level, the average age has risen from 67 years in 2000 to 73 in 2019 [1]. This is mainly due to improved socioeconomic and environmental conditions and better treatments and medical care [2]. Along with the rise in life expectancy, an increase in the volume of chronic diseases has been noted. Patients affected by chronic health conditions often require frequent and specialized medical care that are not easily accessible in rural areas. All these factors increase the overall costs of healthcare services to monitor patients, communicate with them, and offer proper care.

In this framework, e-Health, defined by the World Health Organization as “the cost-effective and secure use of information and communication technologies (ICT) in support of health and health-related fields”[3] may be an essential tool for improving healthcare accessibility while, at the same time, reducing costs. Compared to the concept of telemedicine, which refers to a reduction in the geographic distance between the patient and medical personnel, e-Health is much broader and impacts the whole health system from an organizational perspective [4,5]. Furthermore, according to the WHO, e-Health encompasses “multiple interventions, including telehealth, telemedicine, mobile health (mHealth), electronic medical or health records (eMR/eHR), big data, wearables, and even artificial intelligence” [3]. 

With the outbreak of the COVID-19 pandemic and the consequent introduction of social distancing rules, the use of e-Health to monitor and assist patients has dramatically accelerated [6,7,8,9,10,11,12,13]. Since the very beginning of the pandemic, e-Health and telemedicine tools have been used for forward triage and screening, telemonitoring, infection control procedures, televisits, teleconsultation with experts, and data and report sharing [14]. Later, more solutions were implemented to increase resilience and offer adequate assistance in collecting and sharing data among citizens, healthcare institutions, decision-makers, and public entities engaged in disaster management [15]. 

Several medical specialities have started to implement telemedicine and e-Health solutions following the pandemic, for instance, for neurodegenerative disorders, such as Alzheimer’s disease and amyotrophic lateral sclerosis. As these diseases progress, patients experience a loss of autonomy in daily life activities, becoming more dependent on their caregivers. The outbreak of the COVID-19 pandemic led to the confinement of the majority of the world population at home [7], with even longer periods of isolation recommended for frail patients, thus hindering most chronic/neurodegenerative patients from being assisted in person. In order to assist patients at home, telemedicine solutions were implemented to monitor patients and support them and their caregivers. The post-pandemic literature shows the potential advantages of this kind of monitoring in patient management [16,17,18,19,20].

All in all, the pandemic and the management responses to COVID-19 have sped the adoption of digital technologies in healthcare and beyond [21,22,23,24] by several years [25]. As a result, e-Health has become the third fastest-growing healthcare industry, after pharmaceuticals and medical devices [26], with an increasing number of applications, and even gave birth to the new concept of the Virtual Hospital [27,28,29].

These premises highlight the increasing importance of the e-Health concept, which should be studied more in-depth to evaluate in detail its economic value and general performance. In fact, in the international literature reviews currently available, there seems to be a lack of systematic methods able to analyze these aspects [30,31]. For this reason, several recent research papers have called for further studies and models of analysis on the topic [30,32,33].

Among the most common methods reported by the literature for the economic assessment of e-Health [34], we can mention:The cost-effectiveness analysis (CEA), which compares the costs of a program with its nonmonetary outcomes (e.g., life years gained, diseases avoided) [30,32,35,36,37];The cost–benefit analysis (CBA), which compares the costs and the benefits in monetary terms. Results gathered from this method may later be converted into broader measures of value [32];The cost–utility analysis (CUA), which measures the benefits in terms of utility (e.g., quality-weighted life years gained) through the quality-adjusted life years (QALY) method. This analysis requires a study of direct, indirect, and lost productivity costs [30,32].

Starting from these premises, the aim of this study was to investigate the evolution of the different methods proposed in the literature for the economic and performance evaluation of e-Health, by conducting a structured review of the most recent literature on the topic, especially in a pre- and post-pandemic scenario.

## 2. Materials and Methods

A structured literature review was performed [38] using the Scopus database, the largest dataset of abstracts and citations of peer-reviewed literature in the fields of science, technology, medicine, social sciences, arts, and humanities [39], as well as the datasets Web of Science (WoS) and Pubmed.

A preliminary research protocol was established to document the procedures that were followed in conducting the literature review in order to make it reproducible and reliable (validity). Preliminary research questions were developed to provide new insights [40]. The formalization of the research protocol helped us to identify the central question to be investigated, defined as follows:


*Research Question (RQ): What are the most frequent methods for measuring the economic value and the performance of services in the framework of e-Health in a pre- and post-pandemic scenario?*


The query terms “e-health”, “telemedicine”, “digital health”, and “telehealth” were used in combination with words that represented the three evaluation methods and performance measurement indications. The same search, tailored to the specific search engine characteristics, was conducted on Scopus, WoS, and Pubmed.

The findings from the first step of the analysis, which targeted articles’ titles, abstracts, and keywords, gave 2063 results on Scopus, 5136 on WoS, and 2447 on Pubmed. The search was then further refined by including only journal articles written in English, related to topics in medicine, economics, econometrics, and finance published after 2019, in order to obtain a framework that mirrored the context of the study, also considering the fast development of the latest technologies and the COVID-19 booster effect. The new search led to 439 unique results, excluding duplicates.

Once all the titles and the abstracts were read, 21 papers were finally selected and analyzed by two authors (HB and FD). As a result of this latter step, reading the full text, 20 papers were considered eligible, while one was marked as off-topic. Figure 1 below summarizes the process of analysis and selection of contributions to be included in the sample according to the PRISMA methodology [41,42].

The selected articles were coded and analyzed using Nvivo software (version 12).

The main nodes and subnodes were selected according to the literature [43,44,45] and adapted to the aim of the present study.

The first node collected information on the author’s profile, such as academics, clinicians, and multidisciplinary groups, while the second contained information about the location in which the reported studies were conducted [44,46,47,48,49]. Details about the analyzed sector (public or private) were included in the third node, while information about the applied research methodology was coded in the fourth one [45]. The fifth and sixth nodes collected information about clinical discipline and pathology targeted mentioned by the 20 papers under review [43], while the seventh, eighth, ninth, and tenth nodes reported information on research implications, practices, policies, and outcomes. In the last node, the e-Health tools used were mapped.

## 3. Results

The 20 articles reported in the sample are summarized in the following Table 1, along with their publishing information. 

**Table 1 ijerph-20-04038-t001:** Articles analyzed and coded.

N.	Authors	Title	Year	Journal	Ref.
1	Marcuzzi, A., Bach, K., Nordstoga, A.L., Bertheussen, G.F., Ashikhmin, I., Boldermo, N., Kvarner, E.-N., Nilsen, T.I.L., Marchand, G.H., Ose, S.O., Aasdahl, L., Kaspersen, S.L.,	Individually tailored self-management app-based intervention (selfBACK) versus a self-management web-based intervention (e-Help) or usual care in people with low back and neck pain referred to secondary care: Protocol for a multiarm randomised clinical trial	2021	*BMJ Open*	[50]
2	Buntrock, C., Kählke, F., Smit, F., Ebert, D.D.	A systematic review of trial-based economic evaluations of internet- and mobile-based interventions for substance use disorders	2021	*European Journal of Public Health*	[51]
3	Ochoa-Arnedo, C., Medina, J.C., Flix-Valle, A., Anastasiadou, D.	E-health ecosystem with integrated and stepped psychosocial services for breast cancer survivors: Study protocol of a multicentre randomised controlled trial	2021	*BMJ Open*	[52]
4	Cadilhac, D.A., Sheppard, L., Kim, J., et al.	Economic evaluation protocol and statistical analysis plan for the cost-effectiveness of a novel Australian stroke telemedicine (VST) program	2021	*Frontiers in Neurology*	[53]
5	Ionov, M.V., Zhukova, O.V., Yudina, Y.S., et al.	Value-based approach to blood pressure telemonitoring and remote counseling in hypertensive patients	2021	*Blood Pressure*	[54]
6	Rubee, D., Jinghua, L., Donglan, Z., et al.	An economic evaluation of a mobile text messaging intervention to improve mental health care in resource-poor communities in China: a cost-effectiveness study	2020	*BMC Health Services Research*	[55]
7	Terhorst, Y., Braun, L., Titzler, I., et al.	Clinical and cost-effectiveness of a guided internet-based Acceptance and Commitment Therapy to improve chronic pain–related disability in green professions (PACT-A): study protocol of a pragmatic randomised controlled trial	2020	*BMC*	[56]
8	Fioratti, I., Saragiotto, B.T., Reis, F.J.J., et al.	Evaluation of the efficacy of an internet-based pain education and exercise program for chronic musculoskeletal pain in comparison with online self-management booklet: a protocol of a randomised controlled trial with assessor-blinded, 12-month follow-up, and economic evaluation	2020	*BMC Musculoskeletal Disorders*	[57]
9	Birkemeyer, R., Müller, A., Wahler, S., et al.	A cost-effectiveness analysis model of Preventicus atrial fibrillation screening from the point of view of statutory health insurance in Germany	2020	*Health Economics Review*	[58]
10	Koppenaal, T., Arensman, R.M., van Dongen, J.M., et al.	Effectiveness and cost-effectiveness of stratified blended physiotherapy in patients with non-specific low back pain: study protocol of a cluster randomized controlled trial	2020	*BMC Musculoskeletal Disorders*	[59]
11	Tsou, C., Robinson, S., Boyd, J., et al.	Effectiveness and cost-effectiveness of telehealth in rural and remote emergency departments: a systematic review protocol	2020	*Systematic Reviews*	[60]
12	Yang, Y., Chen, H., Qazi, H., et al.	Intervention and Evaluation of Mobile Health Technologies in Management of Patients Undergoing Chronic Dialysis: Scoping Review	2020	*JMIR mHealth and uHealth*	[61]
13	Thao, V., Nyman, J.A., Nelson, D.B., et al.	Cost-effectiveness of population-level proactive tobacco cessation outreach among socio-economically disadvantaged smokers: evaluation of a randomized control trial	2019	*Addiction*	[62]
14	Nadort, E., Schouten, R.W., Dekker, F.W., et al.	The (cost) effectiveness of guided internet-based self-help CBT for dialysis patients with symptoms of depression: study protocol of a randomised controlled trial	2019	*BMC Psychiatry*	[56]
15	de Ruijter, D., Hoving, C., Evers, S., et al.	An economic evaluation of a computer-tailored e-learning program to promote smoking cessation counseling guideline adherence among practice nurses	2019	*Patient Education and Counseling*	[63]
16	Braun, L., Titzler, I., Ebert, D.D., et al.	Clinical and cost-effectiveness of guided internet-based interventions in the indicated prevention of depression in green professions (PROD-A): study protocol of a 36-month follow-up pragmatic randomized controlled trial	2019	*BMC Psychiatry*	[56]
17	Lizée, T., Basch, E., Trémolières, P., et al.	Cost-Effectiveness of Web-Based Patient-Reported Outcome Surveillance in Patients with Lung Cancer	2019	*Journal of Thoracic Oncology*	[64]
18	Golchert, J., Roehr, S., Berg, F., et al.	HELP@APP: development and evaluation of a self-help app for traumatized Syrian refugees in Germany—a study protocol of a randomized controlled trial	2019	*BMC Psychiatry*	[65]
19	Williams, A., Van Dongen, J.M., Kamper, S.J., et al.	Economic evaluation of a healthy lifestyle intervention for chronic low back pain: A randomized controlled trial	2019	*European Journal of Pain*	[66]
20	Willcox, M., Moorthy, A., Mohan, D., et al.	Mobile Technology for Community Health in Ghana: Is Maternal Messaging and Provider Use of Technology Cost-Effective in Improving Maternal and Child Health Outcomes at Scale?	2019	*Journal of Medical Internet Research*	[67]

A total of 19 out of 20 articles (95% of the sample) were published in medical journals, and only 1 (1 of the 2 systematic reviews) was published in the journal of *Systematic Reviews.*

The average number of authors per paper analyzed was approximately eight. Specifically, papers that were written only by clinicians or academics (5 in total) had, on average, 7.6 authors, which is slightly lower than the number of authors of the papers written by multidisciplinary groups (15 in total), which was 8.4. From this perspective, it is interesting to note that even though the majority of papers were published in medical journals, the authors of the studies were not only clinicians. On the contrary, different competencies were brought together.

Geographically, it is crystal clear that most of the studies were conducted in European countries, specifically in Germany and the Netherlands, where there is a peak of contributions. As for Oceania and America, only two contributions were found, and only one for Africa and Asia continents. Results are reported in Table 2.

More than half of the contributions analyzed are research protocols (11 contributions, 55% of the total sample), which present methods that will later be applied in order to proceed with the performance analysis of programs in the e-Health field. Seven contributions (35% of the total sample) are instead quantitative clinical cases, which, therefore, apply the identified evaluation methods to the practical case. Four of these quantitative clinical cases analyze non-European situations: two Australian, one American, and one African.

In analyzing the content of the various papers, in relation to the clinical discipline and the pathology treated under study, it can be seen that the medical fields most affected are pain medicine and mental health, as represented in Figure 2. 

The most frequently mentioned treated pathologies are pain, such as low back and neck pain, psychological problems, including depression, schizophrenia and post-traumatic stress, and alcohol and smoking addictions, as shown in Figure 3 below.

Considering the suggestions and final remarks, research implications are almost always present, in contrast to practical implications, which are named in only nine contributions (45% of the total sample), and policy implications in eight (40%). The research results are present in the quantitative clinical cases but not present in the protocols and reviews, which instead report state of the art in the field.

The e-Health tools presented in the reviewed articles, or which are used by described programs, vary, but with a higher diffusion of mobile apps, phone calls and online sessions, as illustrated in Figure 4 below.

The methods used for measuring performance in the analyzed e-Health programs are indicated in the identified contributions. Table 3 below provides a description of the measurement methods, placing the data side by side with the treated pathology and e-Health tools used, presenting a visual to help us determine whether there may be links among the three variables. Table 4 aggregates the papers in the sample by pathology, reporting the e-Health tool and method(s) of performance measurement.

Concerning pre- and post-pandemic differences, 12 articles (60% of the sample) were published between 2020 and 2021, while the remaining 8 papers were published in the pre-pandemic era. Still, only one of the papers [54] explicitly mentions the pandemic. In particular, the study by Ionov and colleagues [54] underlines the need to find solutions for patients confined at home. Concerning the tools used, telemonitoring with or without remote counseling may improve blood pressure control and adherence to protocols, especially in force majeure events such as a pandemic outbreak. Still, no specific impact on costs or cost calculation is reported.

## 4. Discussion

Almost all articles published in medical journals express an almost exclusive interest in discussing the topic solely within the boundaries of the clinical setting, presenting a gap in the managerial and statistical literature. This result leads to an interesting reflection, which underlines the increasing relevance of economic and managerial issues in the clinical literature as well. Such findings are consistent with the growing importance of the role of medical doctors as “hybrid managers” [68,69] with cost, budget, and economic outcome responsibilities. Department chiefs should, therefore, be in charge of managing such aspects, which differ from pure clinical practice, and require specific training and decision-making mindsets [70].

The geographical spread of the contributions suggests that the e-Health phenomenon may have recently been widespread in Northern Europe or that it originated in these areas first, allowing for sufficient data to make reasoning about performance evaluations. However, the fact that only four of the seven quantitative clinical cases analyze non-European situations—two Australian, one American, and one African—suggests that it is Europe that is at an earlier stage than the other continents. A few contributions propose field studies that have already been carried out. Europe has thus been the most interested region in the topic for the past four years, but research protocols have a significant impact on the data.

Relative to the clinical discipline and the treated pathology, pain medicine and mental health appear to be the ones attracting the most interest, as described in Figure 2. For this reason, it can be assumed that these disciplines offer more possibilities for remote clinical pathways compared to others, that more often require direct contact with medical personnel [71]. Still, even those specialities that require an in-person approach, such as surgery, allow for online pre-surgical consultation, follow-up, and telemonitoring in the rehabilitation phase [6,72,73,74,75], despite doubts and open questions posed by surgeons about the practical applications and “its efficacy in improving patients’ health, cost-effectiveness and user satisfaction” [72]. 

This reflection is linked to recent studies on the possibility of launching integrated patient management paths according to the Virtual Hospital model [28,29]. This model offers continuous assistance for the patient, carried out remotely, similar to that provided in a physical hospital. A high level of digitalization permits early identification and analysis of diseases, enabling proactive intervention (defined as “initiative medicine”) and thus improving the understanding of disease progression, resulting in a significant reduction in mortality and a substantial improvement in quality of life.

Moreover, in Virtual Hospitals, the number of patients who can be cared for remotely is greater than that in physical hospitals, and this is because patients can be cared for from anywhere (their own home, residences for the elderly, nursing homes or hospices, or other care facilities), without the need for outpatient clinics or hospitals [76,77]. A Virtual Hospital offers numerous advantages due to its unique and high-tech environment, both for patients and healthcare providers as well as for the healthcare institution itself. Furthermore, this model ensures better accessibility and equity of care and healthcare by providing access to services not otherwise available (thus reducing inequalities in access to healthcare services) and offers greater efficiency, especially in monitoring elderly or chronically ill patients who require follow-up care [78]. In fact, it seems to be more efficient if it is aimed at a specific target group of patients, i.e., those who are in a follow-up phase. These include, for example, frail, elderly patients and/or those who have one or more chronic conditions, such as heart disease, stroke, diabetes, chronic respiratory disorders, etc. [79]. Therefore, when applying Virtual Hospital frameworks to such conditions, e-Health tools represent key aspects, and so their economic and sustainability-related issues and performances should be considered and monitored.

Compared to other e-Health tools, the significant diffusion of apps, text messages, and phone calls is undoubtedly due to the fact that these are the most easily used devices for a wide target population. More complex techniques require more specific tools and skills on the part of both the patient and the clinical staff [71], who do not always have adequate training in technology and data analysis [80]. In this regard, one of the major criticalities is a poor level of digital literacy, which affects not only the population (potential patients), but also the health personnel themselves. While the acquisition of new digital skills may be easy for the younger segment of the population and for clinicians—for whom competencies may be implemented during undergraduate or postgraduate modules or in their lifelong learning education—in other cases, training and accompaniment in the use of these tools may be necessary, both among patients and caregivers [81,82]. In some cases, the accompaniment of the patient by a third-party figure (for example, for seniors) may also be necessary. Moreover, the use of complex tools implies the adoption of a new digital mindset by healthcare personnel, patients, and caregivers [71,79]. On the other hand, widely used online tools (such as those related to mobile technology) may represent facilitators in the management of the clinical relationship [83,84,85,86] and in the related communication with the patients.

The recent COVID-19 pandemic, with the reorganization of several clinical processes [10,87,88,89,90], has forced clinicians and clinical institutions to use and apply e-Health tools to monitor patients [11,14], assist them, even in end-of-life care [12], and communicate with them. The COVID-19 experience has, therefore, encouraged and promoted the use of e-Health tools, which have been named among the winning strategies for a resilient and antifragile response to the post-pandemic healthcare system [91]. Still, the results of our literature review do not reveal any particular changes before and after the pandemic outbreak in terms of costs or cost calculation. Interesting enough, only one paper among those published after the beginning of the COVID-19 pandemic specifically mentions this issue [54]. What could be determined was that the pandemic appears to be the ideal context to foster and encourage the use of telemedicine and e-Health tools, but the economic and performance evaluation issue appears independent, as it is discussed by a specific part of the literature. Undoubtedly, the COVID-19 emergency and the subsequent increase in the use of e-Health tools and applications require a deep understanding of the surrounding economic dynamics.

In agreement with other recent studies [30,37], our literature review, albeit it had a limited sample, does not reveal any link or repeatability between the pathology treated and the e-Health tool used, as shown in Table 4. Thus, it might be worthwhile to devote specific studies to determine whether there are more or less suitable or effective instruments relative to each pathology treated at a distance.

## 5. Conclusions

E-Health appears to be a growing phenomenon, especially in the COVID-19 post-pandemic era. E-Health is destined to be one of the winning strategies for caring for an increasing number of patients while controlling healthcare costs. Moreover, it is at the basis of modern phenomena such as the Virtual Hospital [79,92]. Within this context, cost dynamics are relevant, as they require measuring the performance of e-Health tools.

The studies included in the search protocol of this literature review identified a combined use of the three main methods—cost-effectiveness, cost–benefit, and cost–utility—with no preference emerging for any one depending on the pathology identified. Therefore, we identified an effort in the medical literature to understand not only the clinical result but also the economic outcome of the use of e-Health tools linked to new technologies.

Although the sample selected was limited, multiple pathologies and various technological tools for patient support emerged. This fact emphasizes once again the strategic role that e-Health tools are playing in the healthcare landscape and their future development prospects, also from a Virtual Hospital perspective.

The cross-fertilization between economic studies and clinical outcomes appears to be an efficient way to study and understand the phenomenon as a cornerstone for the development of the future health system. Furthermore, in agreement with other literature reviews [30], the need to set standard and shareable guidelines is recalled. In this sense, the role of scientific societies could be strategic in guiding the clinical and managerial community towards certain solutions and methods that are more relevant to specific situations.

### Limitations of the Study and Future Research Avenues

As with every piece of research, our study has several limitations. Although the methodology used to select the literature for the analysis is rigorous and has already been used by multiple international studies, the sample size is far too limited. The studies did not identify a precise link between the pathology, the preferred e-Health tool, and the performance evaluation method. Still, this limitation could be overcome by changing the search keys or adding more specific ones, such as, for instance, “telepsychiatry”, “telecardiology”, or “telephysiotherapy”. A more comprehensive article sample may also allow a comparison of methodologies and technologies applied to the different e-Health types to reveal new practical implications for healthcare institutions and clinicians. Moreover, as e-Health stands as a general topic, which today involves a variety of medical specialities and diseases, more focused research could deepen the same analysis on specific conditions or subjects. 

Finally, given the speed of technological and also organizational change in the healthcare domain also following the effects and responses to the COVID-19 pandemic, it would be appropriate to repeat the investigation in the near future in order to understand innovations and, thus, paradigm shifts in the use and economic measurement of performance. All these aspects constitute interesting future lines of research. Clinicians and experts in economics, healthcare management, and statistics should combine their expertise to produce multidisciplinary results that can help the medical sphere to fully understand, map, and implement the e-Health phenomenon and its potential.

## Figures and Tables

**Figure 1 ijerph-20-04038-f001:**
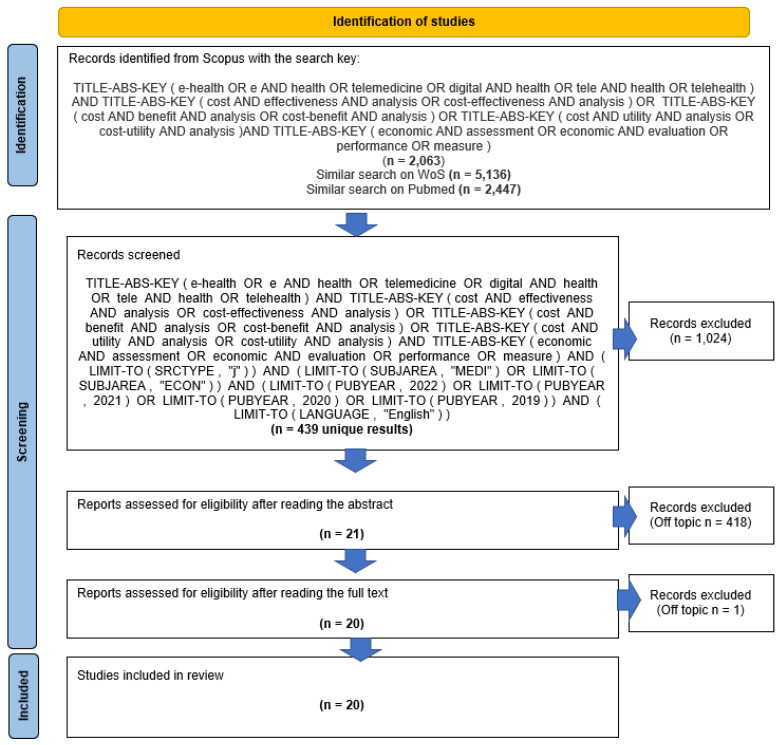
Flowchart of literature review steps, according to the PRISMA protocol. Adapted from Page et al. [42]. Search conducted on 3 December 2022.

**Figure 2 ijerph-20-04038-f002:**
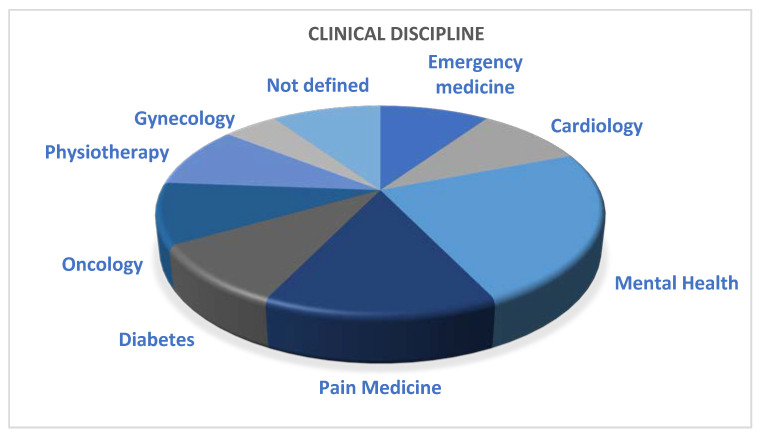
Clinical disciplines covered in the reviewed papers.

**Figure 3 ijerph-20-04038-f003:**
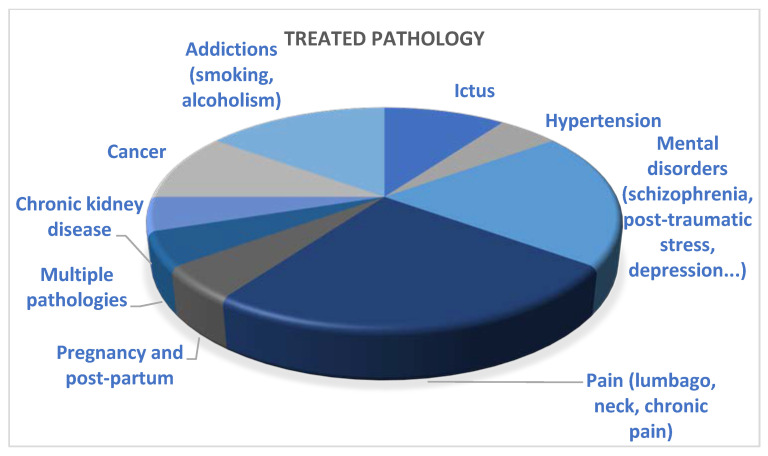
Treated pathologies in the reviewed papers.

**Figure 4 ijerph-20-04038-f004:**
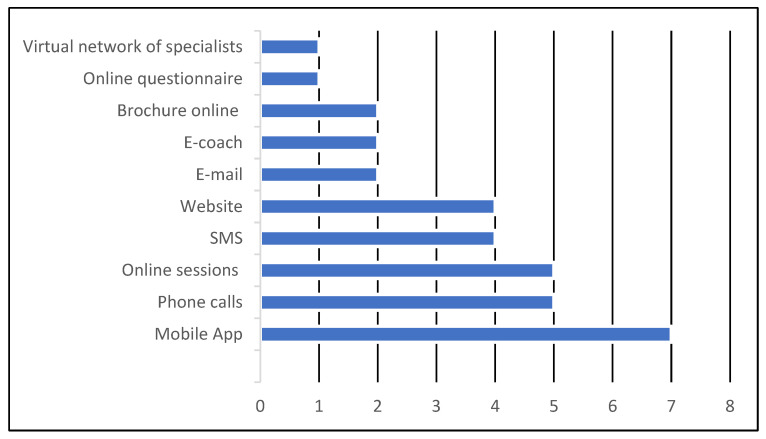
E-Health tools presented in the reviewed articles.

**Table 2 ijerph-20-04038-t002:** Geographical areas of the studies under review.

Continent	Nation	Number of Contributions
Europe	France	1
	Germany	4
	Netherlands	3
	Russia	1
	Norway	1
	Spain	1
America	United States	1
	Brazil	1
Africa	Ghana	1
Oceania	Australia	2
Asia	China	1

**Table 3 ijerph-20-04038-t003:** Performance measurement methods, treated pathology, and tools used for each contribution analyzed (excluding literature reviews).

Title	Treated Pathology	Tools Used	Performance Measurement Methods
Economic evaluation protocol and statistical analysis plan for the cost-effectiveness of a novel Australian stroke telemedicine (VST) program	Ictus	Virtual network of specialists	CEA CBA CUA
Value-based approach to blood pressure telemonitoring and remote counseling in hypertensive patients	Hypertension	App, Website	CUA
An economic evaluation of a mobile text messaging intervention to improve mental health care in resource-poor communities in China: a cost-effectiveness study	Schizophrenia	SMS	CEA CUA
Clinical and cost-effectiveness of a guided internet-based Acceptance and Commitment Therapy to improve chronic pain–related disability in green professions (PACT-A): study protocol of a pragmatic randomised controlled trial	Chronic pain	E-mail, Phone calls, Online sessions, E-coach	CEA CUA
Evaluation of the efficacy of an internet-based pain education and exercise program for chronic musculoskeletal pain in comparison with online self-management booklet: a protocol of a randomised controlled trial with assessor-blinded, 12-month follow-up, and economic evaluation	Chronic pain	SMS, Phone calls, Website, Online brochure	CEA CUA
A cost-effectiveness analysis model of Preventicus atrial fibrillation screening from the point of view of statutory health insurance in Germany	Ictus	App	CEA
Effectiveness and cost-effectiveness of stratified blended physiotherapy in patients with non-specific low back pain: study protocol of a cluster randomized controlled trial	Lumbago	App	CEA
Effectiveness and cost-effectiveness of telehealth in rural and remote emergency departments: a systematic review protocol	Multiple pathologies	Not defined	CEA
Cost-effectiveness of population-level proactive tobacco cessation outreach among socio-economically disadvantaged smokers: evaluation of a randomized control trial	Smoking	E-mail, Phone calls	CEA
The (cost) effectiveness of guided internet-based self-help CBT for dialysis patients with symptoms of depression: study protocol of a randomised controlled trial	Depression	Online sessions	CEA CUA
An economic evaluation of a computer-tailored e-learning program to promote smoking cessation counseling guideline adherence among practice nurses	Smoking	Online sessions	CEA CUA
Clinical and cost-effectiveness of guided internet-based interventions in the indicated prevention of depression in green professions (PROD-A): study protocol of a 36-month follow-up pragmatic randomized controlled trial	Depression	SMS, Phone calls, Online sessions, E-coach	CEA CUA
Cost-Effectiveness of Web-Based Patient-Reported Outcome Surveillance in Patients with Lung Cancer	Lung cancer	Online questionnaire	CEA
HELP@APP: development and evaluation of a self-help app for traumatized Syrian refugees in Germany—a study protocol of a randomized controlled trial	Post-traumatic stress	App	CEA CBA
Economic evaluation of a healthy lifestyle intervention for chronic low back pain: A randomized controlled trial	Lumbago	Phone calls	CEA
Individually tailored self-management app-based intervention (selfBACK) versus a self-management web-based intervention (e-Help) or usual care in people with low back and neck pain referred to secondary care: protocol for a multiarm randomised clinical trial	Lumbago neck pain	App, Website	CEA
E-health ecosystem with integrated and stepped psychosocial services for breast cancer survivors: study protocol of a multicentre randomised controlled trial	Breast cancer	Website	CEA CUA

**Table 4 ijerph-20-04038-t004:** Possible relationships among the treated pathologies, tools, and performance measurement methods used.

Title	Treated Pathology	Tools Used	Performance Measurement Methods
Economic evaluation protocol and statistical analysis plan for the cost-effectiveness of a novel Australian stroke telemedicine (VST) program	Ictus	Virtual network of specialists	CEA CBA CUA
A cost-effectiveness analysis model of Preventicus atrial fibrillation screening from the point of view of statutory health insurance in Germany		App	CEA
An economic evaluation of a mobile text messaging intervention to improve mental health care in resource-poor communities in China: a cost-effectiveness study	Mental disorders	SMS	CEA CUA
The (cost) effectiveness of guided internet-based self-help CBT for dialysis patients with symptoms of depression: study protocol of a randomised controlled trial		Online sessions	CEA CUA
Clinical and cost-effectiveness of guided internet-based interventions in the indicated prevention of depression in green professions (PROD-A): study protocol of a 36-month follow-up pragmatic randomized controlled trial		SMS, Phone calls, Online sessions, E-coach	CEA CUA
HELP@APP: development and evaluation of a self-help app for traumatized Syrian refugees in Germany—a study protocol of a randomized controlled trial		App	CEA CBA
Clinical and cost-effectiveness of a guided internet-based Acceptance and Commitment Therapy to improve chronic pain–related disability in green professions (PACT-A): study protocol of a pragmatic randomised controlled trial	Pain	E-mail, Phone calls, Online sessions, E-coach	CEA CUA
Effectiveness and cost-effectiveness of stratified blended physiotherapy in patients with non-specific low back pain: study protocol of a cluster randomized controlled trial		App	CEA
Evaluation of the efficacy of an internet-based pain education and exercise program for chronic musculoskeletal pain in comparison with online self-management booklet: a protocol of a randomised controlled trial with assessor-blinded, 12-month follow-up, and economic evaluation		SMS, Phone calls, Website, Online brochure	CEA CUA
Economic evaluation of a healthy lifestyle intervention for chronic low back pain: A randomized controlled trial		Phone calls	CEA
Individually tailored self- management app- based intervention (selfBACK) versus a self- management web- based intervention (e-Help) or usual care in people with low back and neck pain referred to secondary care: protocol for a multiarm randomised clinical trial		App, Website	CEA
Cost-effectiveness of population-level proactive tobacco cessation outreach among socio-economically disadvantaged smokers: evaluation of a randomized control trial	Smoking	E-mail, Phone calls	CEA
An economic evaluation of a computer-tailored e-learning program to promote smoking cessation counseling guideline adherence among practice nurses		Online sessions	CEA CUA
Cost-Effectiveness of Web-Based Patient-Reported Outcome Surveillance in Patients with Lung Cancer	Cancer	Online questionnaire	CEA
E- health ecosystem with integrated and stepped psychosocial services for breast cancer survivors: study protocol of a multicentre randomised controlled trial		Website	CEA CUA

## Data Availability

Data can be provided by the corresponding author upon reasonable request.

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
