# Peer review of "Economic and Performance Evaluation of E-Health before and after the Pandemic Era: A Literature Review and Future Perspectives"

_ijerph, 2023, doi:10.3390/ijerph20054038_

Round 1

Reviewer 1 Report

This review paper covers the economic and performance evaluation of e-health.

The paper is in its very preliminary form at the moment. E-health is a hot topic, especially after the pandemic and the number of references should be over hundreds. 

Every section in the paper requires much more detail in it. 

There should be a comparison of methodologies and technologies applied to the different e-health types. 

I would recommend a major revision. 

Author Response

Dear reviewer, 

Please  see the attached file. It reports all the reviewers' responses. Your is highlighted in light blue.

Thanks again for assessing our manuscript and for the valuable comments.

Reviewer 2 Report

The article provides a comprehensive review of the current state of e-health in the aftermath of the COVID-19 pandemic. The authors analyze the economic and performance implications of e-health based on carefully selected sources. The article does a great job of highlighting the potential benefits of e-health, and reduced healthcare costs. The authors also discuss some of the challenges facing e-health, as well as the need for better integration of e-health technologies into existing healthcare systems.

In terms of methodology, the article is well-written and adheres to established standards. The selection of articles for inclusion in the review was conducted using the PRISMA method, ensuring a rigorous and systematic approach. The results of the literature search are presented in a clear and easy-to-read table, making it easy for readers to understand the findings. Overall, the authors have demonstrated a strong attention to detail in their methodology, which adds credibility to the conclusions presented in the article.

I only have a few minor suggestion, which may be helpful to improve the manuscript:

1. The query terms did not include terms specific for different medical specialties like "telepsychiatry", "telecardiology" or "telephysiotherapy".  Do you think, that if you used them, it could have an impact on the number of papers found for this analysis?

2. Figure 1 is not easy to read and it requires some effort to analyze all the details included in it.

3. The papers finally selected for the analysis refer to a few medical specialties like psychiatry, neurology, cardiology, pain medicine, addictions, oncology or hypertensiology. The Authors comment on it in the discussion: "it can be assumed that these are disciplines with more possibilities to offer remote clinical pathways compared to others that more often require direct contact with medical personnel, like surgery." I think it is worth asking some more questions, why surgery is underrepresented in the telemedicine solutions, see e.g. Sartori A, Balla A, Agresta F, Guerrieri M, Ortenzi M. Telemedicine in surgery during COVID-19 pandemic: are we doing enough? Minerva Surg. 2022 Feb;77(1):50-56.

Author Response

(The authors gave the same response as above.)

Reviewer 3 Report

Biancuzzi et al. aimed at exploring the economic value and the role of performance of services in the framework of H-health in the post-pandemic era. Twenty articles were selected among 5000 recent papers on these topics. Most of the articles were published in medical journals, although most articles were written by multi-disciplinary groups, not only including clinicians, but several competencies. Most studies were conducted in Europe. Moreover, 55% of the total sample was represented by research protocols. Clinical disciplines covered were, mainly, mental health, pain medicine, diabetology, emergency medicine, physiotherapy, oncology, and gynecology. The most frequently treated pathologies were low back pain, neck pain, mental disorders, ictus, and then multiple pathologies. The H-health tools principally used were mobile apps, phone calls, and online sessions.  Performances measurements methods were also reviewed. The papers underlines the increasing relevance of economic and managerial issues also for clinicians.

I thank for the opportunity of reviewing this very interesting manuscript. Overall, I think that this paper approaches a relevant topic regarding current need of implementation of e-health. Strength points are: the methodological approach; the readability of the paper (especially of the figures and tables and of the discussion) for multi-disciplinary groups. I have only a minor suggestion.

Among the cited chronic conditions which need an accurate/continuous follow-up, I suggest to consider also neurodegenerative disorders, such as Alzheimer’s disease and Amyotrophic Lateral Sclerosis. As these diseases progress, patients experience a loss of autonomy in daily life activities, becoming more dependent on their caregivers, and move to telemedicine in order to be assisted at home. The breakout of COVID-19 pandemic has confined the majority of the world population at home, thus hindering most chronic/neurodegenerative patients to be assisted in-person. Moreover, patients and their caregivers were more frequently monitored and supported through telemedicine approaches (Vasta et al., 2021 doi: 10.1080/21678421.2020.1820043; De Marchi et al., 2021 doi: 10.1111/ane.13373; Capozzo et al., 2020 doi: 10.1080/21678421.2020.1773502; Sharbafshaaer et al., 2022 doi:10.3389/fpsyt.2022.904841; De Stefano et al., 2021 doi: 10.3390/brainsci12030310), showing potential advantageous aspects of this kind of monitoring in the patients management. Please briefly address also this topic, referring to these additional papers as appropriate.

Author Response

(The authors gave the same response as above.)

Reviewer 4 Report

The main question that address paper is “the economic and performance evaluation of E-Health” In a special context, the post-pandemic era. The main problem that this evaluation to be credible, the time after official pandemic end must be enough large to have some conclusions supported by evidences. This interval must be clearly delimited.

 The evaluation of 20  is somewhat a “thin” approach.

The paper can be relevant in evolution of e-health in critical circumstances, but specific metrics must be used to compare how  was the e-health before pandemic and thereafter. The influence is clearly the difference between. Mostly, the word “economic” rise complex question the is related mainly to costs  and tools to evaluated the cost and what are the results applied to e-health?

The paper evaluates the post-pandemic conclusion the can be a novelty.

The authors should refer more clearly in quantitative terms about costs, because this are in title. Also, the general subject e-health is very large, almost every disease can be somewhat connected with e-health. A more focus on subject that can provide statistic conclusions, e.g. would improve the quality of the paper.

There are Tables that usually have no conclusion, only a evidence of something, e.g. Table 2: Geographical areas analyzed by the studies under review.

The simple counting of number of papers and the subject of them (diseases) doesn’t support quantitatively “the economic and performance evaluation of E-Health”. There is simple situation counted, not scientific  assertion

The references are appropriate, but there is an important missing. How is reflected pandemic literature (e.g., models of COVID) before and post-pandemic in comparison with pandemic period?

There are assertions with no scientific support, e.g., number of authors and geographic distribution pf paper.

Author Response

(The authors gave the same response as above.)

Reviewer 5 Report

Thank you for the opportunity to review this paper by Helena Biancuzzi and colleagues.

The paper aims to investigate the evolution of the different methods proposed in the literature for the economic and performance evaluation of e-Health and to understand if there is a link between the proposed method of analysis and the e-Health tool used. A structured literature review was performed using the Scopus, Web of Science and Pubmed databases. A preliminary research protocol was established to document the procedures for conducting the literature review in order to make it reproducible and reliable. After the selection, 20 selected articles were coded and analyzed using Nvivo software.

I have some observations related to the quality of the paper:

·         Although there is a lot of talk in the paper about Covid 19/post Covid and the implications in e-health, only 5 of the 20 selected articles are from 2021 and none refer to this topic.

·      The limitations presented by the authors show that the study does not reach its objectives.         

·   Keywords were not well chosen, and the research results are not conclusive.

For these reasons, I am not convinced that the paper is acceptable for publication in IJERPH. I would recommend the rejection of the manuscript.

I hope my feedback is useful to the authors and wish them all the best in pursuing this important area of research.

Author Response

(The authors gave the same response as above.)

Round 2

Reviewer 1 Report

Thanks for working on the paper. It is much better now. 

Author Response

Dear reviewer, thank you for assessing the new version of our manuscript.

Warm regards, 

The Authors

Reviewer 5 Report

Changing the title of the paper makes the 20 papers, selected for being analyzed during the review process, suitable for the mentioned period.

Even if the situation presented in the field covers, by introducing new references, several aspects related to e-health in the pre and post Covid-19 period, still only the same 20 articles are finally analyzed.

In the current format, no important changes are made compared to the initial manuscript, but only justifications for not achieving the objectives of the review and not covering several aspects related to the addressed topic.

It is possible that a research fails. In this case, the research consists in analyzing, with certain established objectives, a number of articles. If, during the analysis, it becomes clear that the result will be a failure, it is necessary to stop the process and modify, in due time, the selection criteria/keywords.

I don't think that a Q1 journal, such as EJERPH, is suited for presenting papers that do not achieve their research goals.

For these reasons, I am not convinced that the paper is acceptable for publication in IJERPH. I would recommend rejection of the manuscript.

Author Response

(The authors gave the same response as above.)
